# Repeated Sprint Performance and Inter-Limb Asymmetry in Elite Female Sprinters: A Study of Lactate Dynamics and Lower Limb Muscle Activity

**DOI:** 10.3390/jfmk10020213

**Published:** 2025-06-04

**Authors:** Artur Gołaś, Artur Terbalyan, Mariola Gepfert, Robert Roczniok, Aleskander Matusiński, Krzysztof Kotuła, Przemysław Pietraszewski, Adam Zając

**Affiliations:** Institute of Sport Sciences, Jerzy Kukuczka Academy of Physical Education, 40-065 Katowice, Poland; a.golas@awf.katowice.pl (A.G.); a.terbalyan@awf.katowice.pl (A.T.); m.gepfert@awf.katowice.pl (M.G.); r.roczniok@awf.katowice.pl (R.R.); a.matusinski@awf.katowice.pl (A.M.); k.kotula@awf.katowice.pl (K.K.); p.pietraszewski@awf.katowice.pl (P.P.)

**Keywords:** anaerobic performance, glycolysis, national team, movement pattern, track and field

## Abstract

**Background:** Repeated short maximal sprints induce high blood lactate concentrations and may influence neuromuscular coordination, but the relationship between lactate accumulation and inter-limb asymmetry in elite athletes remains unclear. **Objectives:** This study sought to investigate how blood lactate dynamics during repeated sprint efforts relate to sprint performance and inter-limb muscle asymmetry in elite female sprinters. **Methods:** Eight elite women (21.3 ± 5.4 y; 54.2 ± 5.1 kg; 165.4 ± 4.3 cm) performed four sets of five all-out 50 m sprints (1 min rest between reps; 5 min between sets). Sprint times were electronically timed. Capillary lactate was measured at rest and 1 min/4 min post-set. Quadricep, hamstring, and gluteal asymmetry was assessed via textile electromyography. Nonparametric statistics (Spearman’s ρ, Friedman test) were applied. **Results:** From Set 1 to Set 4, sprint time fell from 7.25 ± 0.13 s to 7.07 ± 0.16 s (*p* = 0.044), 1 min lactate rose from 8.51 ± 2.65 to 15.60 ± 2.66 mmol/L (*p* < 0.001), and 4 min lactate from 7.16 ± 2.32 to 13.64 ± 2.76 mmol/L (*p* = 0.002). Muscle group asymmetries decreased (*p* < 0.01), with lactate at 1 min, correlating negatively with quadriceps (ρ = −0.59) and gluteal (ρ = −0.48) asymmetry. **Conclusions:** In elite female sprinters, progressive lactate accumulation during repeated 50 m bouts is linked to faster times and reduced muscle asymmetry, suggesting that lactate may enhance neuromuscular coordination under fatigue.

## 1. Introduction

Sprinting over short distances such as 100, 200, and 400 m is characterized by the athlete’s ability to manifest a high level of power output in the shortest possible time in order to reach the finish line as quickly as possible [1]. Repeated sprint training (RST) is a well-known method used by track and field or strength and conditioning practitioners to induce specific adaptive changes that allow athletes to maintain pace over the entire distance of a race and thus optimize their running strategy which involves performing sets of full-effort bouts of sprints without sufficient recovery [2].

The repetitive nature of RST is characterized by specific physiological feedback in the face of lactate accumulation [3]—the byproduct of anaerobic glycolysis, which increases its concentration dramatically when performing multiple sprints with increased work-to-rest ratio in line with progression in training load [4]. The physiological effects of lactate extend far beyond a mere role as a “bad guy”; in the fatigue-inducing molecular cascade, its release triggers unique systemic pathways that affect almost every organ and tissue [5]. In addition, lactate affects several homeostatic functions. This, in turn, affects the “homeostatic tone” of the nervous system [6], which is an area of interest to sports scientists seeking new applications of glycolysis-reliant training programs like RST for enhancing athletic performance.

RST causes severe central and peripheral fatigue inducing a progressive disruption of muscle activation patterns [7], leading to interlimb asymmetries that result in an uneven stride—an issue that has recently been actively debated in relation to sprint performance [8]. Muscle asymmetries are common in running sports due to bilateral force distribution during the dash because of the bipedal nature of this form of locomotion [1]. While asymmetry per se seems to not be associated with improved sprint times [9], it may mediate a long-term decline in athletic performance through the development of injury. Inter-limb asymmetry exceeding 15% [10] has been demonstrated to elevate the risk of injuries. Asymmetries below 9% indicate that the muscle groups on the right and left legs are well balanced. A difference in muscle activity between the right and left limb or side of the body between 9 and 18% indicates a trend towards asymmetry. Asymmetry exceeding 18% indicates imbalances, necessitating corrective measures to prevent injury [11].

Electromyography (EMG) appears to be a reliable tool which has been used successfully in our laboratory in previous research to assess muscle asymmetry during different sprint events [12]. Surface electromyography (sEMG) using functional underwear or portable EMG systems allows the dynamics of changes in muscle activity patterns to be tracked over time [13]. EMG as an asymmetry assessment tool stands as a useful alternative in field research and training process to methods of kinetic and kinematic testing of running mechanics such as a limb symmetry index or asymmetry ratios between limbs or symmetry angle [14], whose applicability is more complicated and time-consuming.

Although there is a growing body of evidence concerning the systemic effects of lactate and energy homeostasis, there is still a lack of data regarding the role of lactate in motor performance. It seems that athletes at the elite level are capable of achieving and sustaining elevated lactate concentrations for longer periods of time than their less advanced counterparts, which enables them to maintain or even improve performance during high-intensity exercise [15]. Therefore, the objective of this study was to determine the relationship between incremental training loads, lactate concentration, sprint performance, and lower limb muscle symmetry. It was hypothesized that enhancements in sprint performance and compensation of muscle asymmetry are aligned with lactate accumulation.

## 2. Materials and Methods

### 2.1. Partipiciants

Eight elite 200–400 m female sprinters, members of the national team, participated in the study. The research was carried out during the competitive 2023 indoor season held at the National Training Center. The participants performed a light technical training session 48 h prior to the intervention and testing to avoid fatigue. Participants were instructed to abstain from alcohol and energy drinks for 48 h and from caffeine on the day of testing. High-intensity exercise was avoided for 48 h prior to each session, and habitual diet and sleep patterns were maintained in the 24 h preceding the test. All athletes consumed a light, familiar meal at least 3 h before testing and confirmed euhydration upon arrival. The athletes were not allowed to use any supplements or stimulants for 48 h before and during the studies. Inclusion criteria required athletes to be female sprinters specializing in the 200–400 m event, currently competing at national or international level, aged 18–30 years, and possessing a minimum of five years of sprint training experience. All participants were free from musculoskeletal injury for at least six months prior to the study and provided written informed consent. Exclusion criteria included any history of cardiovascular, metabolic, or neurological disorders; use of performance-enhancing substances or medications affecting muscle metabolism; and failure to comply with pre-test guidelines (e.g., recent alcohol or stimulant intake, insufficient rest). The study received the approval of the Bioethical Committee of the Academy of Physical Education in Katowice (10/2018) and was carried out according to the ethical standards of the Declaration of Helsinki, 2013.

### 2.2. Study Design

Following a pre-competitive warm-up the female sprinters performed a high intensity interval intervention consisting of 4 sets of 5 repetitions of all out 50 m sprints with a rest interval of 1 min between repetition and 5 min between sets. The intervention was similar in volume and intensity to regular training routines during the competitive season. The athletes performed each sprint from a crouched start, decelerated after the 50 m line and walked back for recovery. Each 50 m distance was timed electronically with the use of Witty Photocells (Witty, Microgate, Bolzano, Italy). Study protocol is illustrated in Figure 1 below.

### 2.3. Lactate Concentration Measurement

Blood lactate samples were drawn from the fingertip of the participant’s non-dominant hand at rest and at both the 1st and 4th minute of recovery following each sprint set. A sterile, single-use lancet was used to puncture the fingertip, and approximately 20 µL of blood was collected into heparinized capillary tubes (75 µL capacity) by capillary action to prevent coagulation. A 0.5 µL aliquot of this blood was immediately transferred onto a test strip for analysis with the Lactate Scout analyzer (SensLab GmbH, Leipzig, Germany). The Lactate Scout employs an enzymatic–amperometric method in which lactate oxidase catalyzes the oxidation of lactate; electrons released during this reaction are shuttled via a mediator to a working electrode, producing a current proportional to lactate concentration. Test strips fill directly from the sample site, and each batch carries a unique calibration code. The device requires ~15 s per measurement and was calibrated before each testing session using the manufacturer’s 1-level calibration solution (9.5–12.5 mmol/L) to ensure accuracy. All measurements were conducted at ambient laboratory temperature (22–24 °C), and any error readings were immediately repeated using fresh capillaries and strips.

### 2.4. Textile Electrodes

Muscle activation was measured using EMG shorts (Myonear Pro, Myontec, Finland), which incorporate conductive electrodes and wires directly into the fabric. These electrodes are positioned over three major muscle groups: the glutei, quadriceps, and hamstrings. Three sizes of shorts (S, M, L) were available, and the best-fitting size was selected for each participant. Proper sizing was essential to ensure good electrode-to-skin contact, minimizing movement artifacts during dynamic movement. A small amount of water was applied to the electrodes prior to wearing the shorts to enhance signal transmission, as recommended by previous research [16]. The EMG signals were measured in a raw form with a sampling frequency of 1000 Hz and a frequency band of 50 Hz–200 Hz (−3 dB). The EMG signals were wirelessly transmitted to a laptop and recorded at a sampling rate of 1000 Hz using MegaWin v 3.0.1 software (Mega Electronics Ltd., Kuopio, Finland). Myoelectric activity was collected during all trials. The data from the right and left leg were included in the subsequent analysis. Finni et al. [16] showed that the signals from the textile electrodes are in good agreement with the traditionally measured surface EMG signal. The relationship between muscle strength and EMG was similar for both traditional and textile electrodes.

Inter-limb EMG asymmetry was quantified for each muscle by calculating the absolute difference between the left and right leg’s EMG activity, normalized to their average, expressed as a percentage. The asymmetry index was computed as:$$Asymmetry\left(\%\right)=\frac{\left|{RMS}_{left}-{RMS}_{right}\right|}{\left({RMS}_{left}+{RMS}_{right}\right)/2}\times 100$$
this yields 0% if the two limbs have equal EMG output and larger percentages for greater asymmetries. RMS—root mean square.

### 2.5. Statistical Analysis

All variables were first tested for normality using the Shapiro–Wilk test; none met the assumption of normality (*p* < 0.05), so data are presented as median (minimum–maximum). Within-condition (repeated-measures) differences were evaluated using Friedman test. Significant Friedman omnibus effects were followed by Durbin–Conover pairwise comparisons; *p*-values were adjusted for multiple testing with the Holm–Bonferroni procedure to control the family-wise error rate (Type I). Associations between variables (e.g., lactate concentration, sprint time, and EMG asymmetry) were assessed using Spearman’s rank correlation coefficient. Statistical analyses and visualizations were performed in Statistica 13.1 (StatSoft, Tulsa, OK, USA) and GraphPad Prism 9.4.1 (GraphPad Software, San Diego, CA, USA). Statistical significance was defined at α = 0.05.

## 3. Results

The average age, body mass and body height were 21.3 ± 5.4 years, 54.2 ± 5.1 kg, and 165.4 ± 4.3 cm, respectively. The athletes had at least 5 years of training and competition experience. Basic descriptive characteristics are presented below in Table 1.

Most of the variables failed the Shapiro–Wilk standard of *p* < 0.05. For the remaining variables, a considerable degree of heterogeneity exceeding V > 20% was also identified. Consequently, non-parametric tests were employed for subsequent analyses. Spearman’s rank correlation coefficient was employed to determine whether each consecutive set of sprints was associated with the dependent variables.

The progressive-sprint protocol elicited marked improvements in both performance and neuromuscular symmetry that were closely linked to the accumulation of metabolic by-products during recovery (Figure 2). Specifically, 50 m sprint times decreased across successive sets (R = −0.50, *p* = 0.034), while blood lactate concentration rose markedly in the first (R = 0.74, *p* < 0.00001) and fourth (R = 0.68, *p* = 0.00002) minutes of each rest interval. Concurrently, interlimb asymmetry diminished for all muscle groups assessed: quadriceps (R = −0.46, *p* = 0.0081), hamstrings (R = −0.71, *p* < 0.001), and gluteus (R = −0.81, *p* < 0.001). Further correlational analyses revealed that higher lactate levels at minutes 1 and 4 of recovery were significantly associated with reduced quadricep (R = −0.59; *p* = 0.0003 and R = −0.46; *p* = 0.0074, respectively) and gluteus asymmetry (R = −0.48; *p* = 0.006 and R = −0.43; *p* = 0.013, respectively).

Significant main effects across sets were found for sprint time (χ^2^(3) = 18.50, *p* < 0.001). Comparisons between Set 4 and Set 1 indicated faster sprint times (Z = 8.18, *p* < 0.001) (Figure 3).

Significant main effects across sets were found for lactate at 1 min (χ^2^(3) = 24.00, *p* < 0.001) and lactate at 4 min (χ^2^(3) = 22.60, *p* < 0.001). Lactate concentration at 1 min post-exercise was significantly higher in Set 4 than in Set 1 (Z = inf, *p* < 0.001) and Set 2 (Z = inf, *p* < 0.001). Likewise, lactate at 4 min post-exercise was elevated in Set 4 compared with Set 1 (Z = 17.70, *p* < 0.001) (Figure 4).

Significant main effects across sets were found for hamstring asymmetry (χ^2^(3) = 24.00, *p* < 0.001), gluteus asymmetry (H(3) = 20.47, *p* = 0.0001), and quadricep asymmetry (χ^2^(3) = 10.20, *p* = 0.017). Hamstring asymmetry decreased progressively, with lower values in Set 4 compared to Set 1 (Z = inf, *p* < 0.001) and Set 2 (Z = inf, *p* < 0.001). Gluteus asymmetry followed a similar pattern, showing significant reductions in Set 4 versus Set 1 (Z = inf, *p* < 0.001), Set 4 versus Set 2 (Z = inf, *p* < 0.001), and Set 3 versus Set 1 (Z = inf, *p* < 0.001). No significant contrasts were observed in quadricep asymmetry across the four sets. These results are illustrated in Figure 5.

## 4. Discussion

This study was designed to determine the relationship between repeated high-intensity exercise, lactate accumulation, sprinting performance, and lower limb muscle asymmetry in elite female sprinters.

Contrary to the common expectation that fatigue invariably degrades performance and coordination, our elite sprinters ran faster and exhibited more balanced muscle activation over successive 50 m sprint sets, even as blood lactate levels rose sharply. This pattern suggests that, at the highest levels of training, athletes may engage neuromuscular or pacing strategies that compensate for metabolic stress—perhaps by optimizing motor unit recruitment or by using moderate lactate accumulation as a temporary energy reservoir—thereby preserving or enhancing sprint mechanics under brief recovery conditions.

These findings add nuance to the literature on fatigue; whereas many studies report performance decrements during repeated-sprint protocols, our data align with emerging evidence that well-trained athletes can maintain, or even improve, output when familiarized with the task and sufficiently warmed up. From a practical standpoint, incorporating progressive all-out sprint sets into training may both prime the neuromuscular system and expose athletes to controlled fatigue, potentially bolstering their ability to sustain coordination in competition.

These results indicate that as lactate accumulation increased, sprint performance improved and muscle asymmetry decreased, supporting the hypothesis of a close relationship between these variables. Furthermore, the reduction in asymmetry with increasing lactate concentration points to a compensatory mechanism in muscle coordination, which may contribute to sustained or improved performance over repeated sprints. There is no similar research available in the scientific literature; however, Bishop et al. [17] assessed interlimb asymmetries after using an identical protocol of four sets of six repetitions of the 40 m sprint in an amateur population. A single-leg CMJ was used as an asymmetry monitoring test. Interlimb asymmetry was greater from set to set, and sprint performance was progressively reduced. However, statistically significant levels of fatigue were observed between all sets with the exception of sets 3 and 4, while the difference in performance between the sets began to decrease; it is possible that increasing training volume could further reduce this change. Such a difference in outcome may be initiated by differences in mastery level or resting times [18,19,20]. Regrettably, lactate concentrations were not evaluated to make a direct comparison.

One potential explanation for the observed association between rising lactate levels (from ~9.0 to ~16.0 mmol/L) and improved coordination is that lactate may act as more than a mere metabolic by-product. Lactate could serve as an alternative fuel for neurons via monocarboxylate transporters, supporting central motor drive during repeated sprints [20,21,22]. It is also conceivable that elevated lactate might enhance motor cortex excitability and promote BDNF release, thereby facilitating synaptic plasticity and more efficient muscle activation patterns [23,24,25,26,27,28,29,30,31]. Furthermore, exercise-induced upregulation of MCT2 in the cerebellum [32,33,34,35,36] and activation of GPR81 receptors in sensorimotor regions [37,38] may represent additional pathways through which lactate influences neuromuscular control. While these mechanisms provide a coherent framework for linking lactate dynamics to reduced EMG asymmetry, direct neurophysiological and molecular studies are needed to test whether lactate causally contributes to enhanced coordination under fatigue in elite sprinters.

Lacour et al. [39] previously showed that athletes reached an extreme state of fatigue by reaching lactate concentrations of up to 20 mmol/L. The subjects in our study were not able to reach such lactate concentrations, most likely because of the exercise protocol, which included short, high-intensity exercise efforts with incomplete rest intervals in between. Extreme acidosis and lactate concentrations above 20 mmol/L have been observed in prolonged sprints lasting 40–50 s. This may explain why in our study the fatigue did not outweigh the ability to enhance coordination and reduce muscular asymmetries. In contrast, the aforementioned study primarily involved male participants at the elite level. While there is no discernible gender difference in the achievement of high lactate concentrations during an all-out effort dash [40], it is possible that a set of gender-specific phenotypic traits may underpin a differential somatosensory response to lactate concentration during exercise. This represents a compelling focus for further research in this area. Furthermore, it would be beneficial to investigate the interrelationship between the studied variables at lactate concentrations exceeding 20 mmol/L.

### Limitations

This study has several important limitations. Firstly, although participants completed a standardized pre-test warm-up (10 min light jog, dynamic stretching, and three progressive sprint accelerations), the observed progressive improvement in sprint times across sets suggests that this routine may not have fully prepared athletes for maximal effort in Set 1. In practice, pre-competition warm-ups are often more intensive—incorporating additional dynamic drills and sprint-specific accelerations—and a more vigorous protocol may be required to ensure true “fresh” performance. Consequently, some of the between-set performance gains and reductions in muscle asymmetry seen here may reflect an extended warm-up effect rather than true adaptation to fatigue and should therefore be interpreted with caution.

Secondly, the sample size was small (N = 8), which limits the statistical power to detect more subtle effects and increases the risk of Type II error. This constraint is common in elite sport research, where the pool of athletes meeting strict inclusion criteria is inherently limited. Recruiting a larger cohort of national- or international-level sprinters was not feasible given training schedules, competition cycles, and the small population of athletes at this performance level. As a result, while these findings offer valuable insights into elite sprint dynamics, they may not be generalizable to larger or more heterogeneous athlete groups.

Finally, focusing on a highly specialized cohort of elite sprinters restricts the external validity of the results. Athletes of different disciplines, performance levels, or training backgrounds may respond differently to repeated-sprint protocols and exhibit distinct lactate–coordination relationships. Future research should replicate these findings in larger samples and across varied athletic populations to determine the broader applicability of these associations.

## 5. Conclusions

In conclusion, during four sets of five all-out 50 m sprints with 1 min rest between repetitions and 5 min between sets, elite sprinters demonstrated progressive improvements in sprint time alongside reductions in inter-limb EMG asymmetry, concurrent with rising blood lactate levels. These results show that lactate accumulation is associated with maintained—or even enhanced—performance and more balanced muscle activation under brief recovery conditions. However, given the correlational design, causality cannot be inferred, and the extent to which lactate directly influences neuromuscular coordination remains to be determined. Future studies should investigate the mechanisms underpinning these associations and assess whether similar patterns occur in other athlete populations or with alternative sprint protocols.

## Figures and Tables

**Figure 1 jfmk-10-00213-f001:**
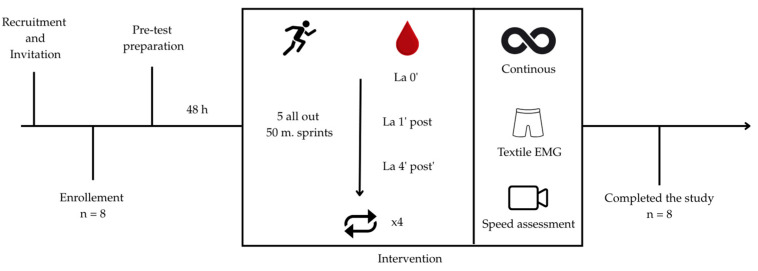
Study protocol.

**Figure 2 jfmk-10-00213-f002:**
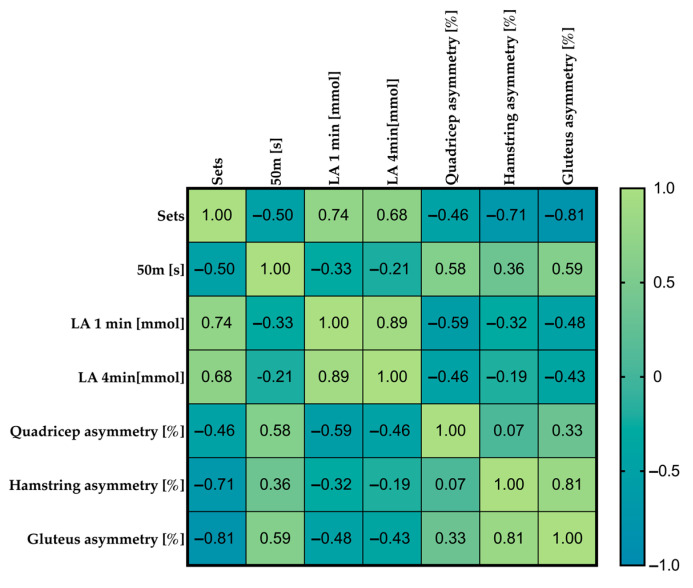
Spearman rank order correlation between the variables of interest.

**Figure 3 jfmk-10-00213-f003:**
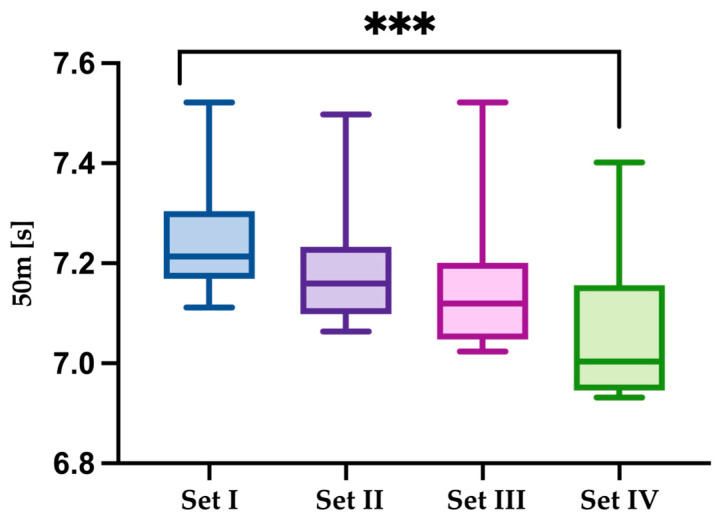
Comparison of results for the dependent variable 50 m [s] for the analyzed sets groups. Statistically significant differences are indicated by asterisks as follows: *** *p* < 0.001.

**Figure 4 jfmk-10-00213-f004:**
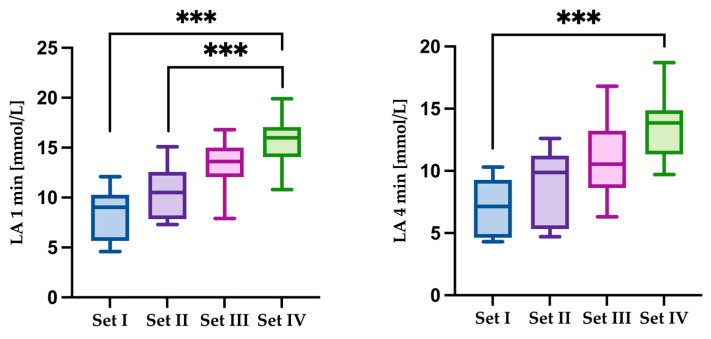
Comparison of results for the dependent variable LA1st min [mmol] and LA4th min of recovery for the considered sets of sprints. Statistically significant differences are indicated by asterisks as follows: *** *p* < 0.001.

**Figure 5 jfmk-10-00213-f005:**
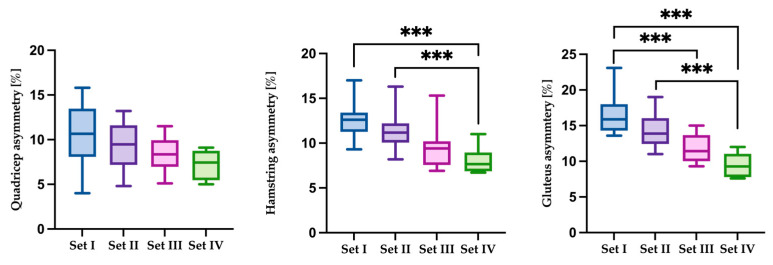
Comparison of results for the dependent variable quadricep, hamstring, and gluteus asymmetry [%] for the analyzed sets of sprints. Statistically significant differences are indicated by asterisks as follows: *** *p* < 0.001.

**Table 1 jfmk-10-00213-t001:** Sprint performance, blood lactate, and EMG asymmetry across four sprint sets.

Variable	Set 1	Set 2	Set 3	Set 4
50 m sprint time (s)	7.21 (7.11–7.52) ^a^	7.16 (7.06–7.50)	7.12 (7.02–7.52)	7.00 (6.93–7.40) ^a^
Lactate, 1 min post-set (mmol/L)	9.0 (4.6–12.1) ^a^	10.5 (7.3–15.1) ^b^	13.6 (7.9–16.8)	16.0 (10.8–19.9) ^a,b^
Lactate, 4 min post-set (mmol/L)	7.1 (4.3–10.3) ^a^	9.9 (4.7–12.6)	10.5 (6.3–16.8)	13.9 (9.7–18.7) ^a^
Quadriceps asymmetry (%)	10.7 (4.0–15.8)	9.5 (4.8–13.2)	8.4 (5.1–11.5)	7.4 (5.0–9.1)
Hamstrings asymmetry (%)	12.6 (9.3–17.0) ^a^	11.2 (8.2–16.3) ^b^	9.4 (6.9–15.3)	7.7 (6.7–11.0) ^a,b^
Gluteus asymmetry (%)	15.9 (13.6–23.1) ^a,c^	13.9 (11.0–19.0) ^b^	11.4 (9.3–15.0) ^c^	9.3 (7.6–12.0) ^a,b^

Values are median (minimum–maximum); ^a, b, c^—significant pairwise comparisons with *p* < 0.05.

## Data Availability

The original contributions presented in this study are included in the article. Further inquiries can be directed to the corresponding author.

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
