# Peer review of "Repeated Sprint Performance and Inter-Limb Asymmetry in Elite Female Sprinters: A Study of Lactate Dynamics and Lower Limb Muscle Activity"

_jfmk, 2025, doi:10.3390/jfmk10020213_

Round 1

Reviewer 1 Report

Comments and Suggestions for Authors

This manuscript addresses an important and challenging topic: the physiological and neuromuscular responses during repeated sprint training in elite female sprinters. The authors should be commended for their effort to conduct research with elite athletes, which is logistically and practically demanding. The inclusion of EMG-based asymmetry analysis and lactate measurements adds a potentially valuable dimension to the understanding of sprint performance. However, several critical issues limit the interpretability and scientific contribution of the current version of the manuscript.

Major Comments

  1. Overinterpretation of Results
    The main conclusion—that improved sprint performance and reduced muscle asymmetry are due to increased lactate concentration—appears speculative and not sufficiently supported by the data. While sprint times decreased and lactate increased across sets, causality cannot be inferred from this correlation. The authors are encouraged to rephrase these statements and clarify whether a theoretical framework supports this interpretation. Are there prior studies that demonstrate enhanced coordination or speed with elevated lactate levels? If so, these should be cited and critically discussed.

  2. Contradiction with Established Knowledge
    It is well established that fatigue typically impairs coordination. The observed reduction in asymmetry with increasing fatigue contradicts this and warrants a more thorough discussion. Why would coordination improve under fatigue in this context? Could this be due to insufficient warm-up or pacing strategies?

  3. Within-Set Analysis
    The manuscript lacks detail on performance and asymmetry trends within each set. Was the same pattern of improvement observed within sets as across sets? This information could provide valuable insight into the dynamics of fatigue and adaptation.

  4. EMG Asymmetry Calculation
    The method for calculating muscle asymmetry from EMG signals is not clearly described. This is a critical methodological detail that must be included in the Methods section to ensure reproducibility and interpretability.

  5. Warm-Up Protocol
    The progressive improvement in sprint times suggests that the pre-competition warm-up may have been insufficient. This should be acknowledged and discussed as a potential confounding factor.

  6. Statistical Analysis
    It is unclear whether corrections for multiple comparisons were applied. Given the number of variables and comparisons, this is essential to control for Type I error.

  7. Discussion Section Structure
    The beginning of the Discussion section repeats numerical results rather than interpreting them. This section should focus on contextualizing the findings, discussing implications, and acknowledging limitations.

  8. Speculative Statements
    Line 260 includes speculative content not directly supported by the data. Such statements should be removed or clearly framed as hypotheses for future research.

  9. Lactate Discussion (L287–375)
    The discussion on lactate physiology is not well integrated with the study’s aims or findings. It appears disconnected and does not help explain the observed results. This section should be revised or significantly shortened.

  10. Conclusions
    The conclusions are largely based on assumptions and literature rather than the presented data. They should be rewritten to reflect only what the data can support.

Specific Comments

  • Table 1: Please round values to meaningful precision (e.g., 10.492 ± 3.668 → 10.5 ± 3.7).
  • Figure 2: Appears to duplicate data from Table 1. Please avoid redundant presentation.
  • Line 98–100: While the statement is true, it is not appropriate in this section and should be relocated or removed.

While the study addresses a relevant topic and involves a valuable population, the current manuscript requires substantial revision in terms of methodology, interpretation, and structure. I encourage the authors to address these concerns and consider resubmission after a thorough revision.

Author Response

We attach the file below. Text changes are highlighted in the manuscript file

Reviewer 2 Report

Comments and Suggestions for Authors

The manuscript entitled “Repeated sprint performance and inter-limb asymmetry in elite sprinters: a study of lactate dynamics and lower limb muscle activity” was reviewed. The article provides interesting information on the topic; however, adjustments need to be made so that the article can continue the path to publication.

I kindly ask if all changes made to the text be highlighted in yellow or a different color in the text.

Below are the reviewer's considerations to be adjusted in the manuscript.

Abstract:

1- Please start the summary with the objectives or separate the “Background” topics from the “Objectives” topics.

2- Restructure the objectives to be closer to the title. In the objectives, you provide information about the type of athlete and gender. The title is very vague.

3- What does the acronym “EMG” mean? Whenever you present an acronym for the first time, present the name in full.

4- You mentioned that nonparametric statistics were performed but presented the data as mean and standard deviation. When the data are not normal, they should be presented as median and interquartile range or minimum and maximum. Please adjust and explain this in the methodology.

Keywords:

5- Avoid repeating the words presented in the title. Replace them with synonyms.

Introduction:

6- The introduction lacks references on the molecular, metabolic mechanisms and cellular adaptations that explain the effects of Sprinting in relation to the variables investigated.

7- The objectives presented at the end of the introduction are different from those presented in the abstract, please standardize.

Materials and Methods:

8- Did you register the study as a clinical trial? I couldn't find the registration number or the website link.

9- Did you perform a sample calculation for the study?

10- Provide information on the equipment used to evaluate the athletes. Brand, manufacturer and technical measurement information.

11- What were the inclusion and exclusion criteria for participants?

12- Create a flowchart for the study. Provide information on how many people were invited and how many completed the study.

13- What precautions did you take with the athletes before collecting lactate? For example, did you ask them to avoid consuming alcoholic beverages 48 hours before the tests, energy drinks and intense physical exercise?

14- Explain in more detail how the blood samples were processed for lactate analysis. For example, were the capillaries heparinized? How much blood was collected per capillary? Among other details. 15- Rewrite the statistical analyses according to the chronological order of the facts. Start by submitting the data to the normality test, then presenting the data (since they were non-parametric, they cannot be presented as mean and standard deviation, but rather as median and interquartile range or minimum and maximum). Discuss the statistical tests that the data will be subjected to and at the end, mention the statistical package used and the significance index.

Results:

16- Start the presentation of the results with the sociodemographic, anthropometric and body composition information of the athletes. Move this information from the methodology to the results.

17- In Table 1, please do not present nonparametric data such as mean and standard deviation. Take the opportunity to describe the main information in Table 1 in text. This makes it easier for the reader to have a focused look at the table, allowing them to extract other information they find relevant.

18- The title of Table 1 is focusing on statistics. Change the title to the name of the main variables that will be presented. The statistical issues have already been mentioned in the specific topic. Take the opportunity to mark with an asterisk where statistical differences occurred and mention it in the table caption.

19- To make the manuscript more attractive to readers, replace Table 2 with a figure and follow the same suggestion presented previously, not naming the table or figure with the statistical analyses performed but with the name of the variables investigated. Furthermore, in Table 2, it is not clear what you are correlating. Use these suggestions for Table 3 as well.

20- When describing the results, there is no need to repeat the names of the statistical tests; simply cite the results of the analyses with their respective p-values.

21- You mention in the text that there are differences between set 1 and set 4 in Figure 1, but you do not indicate these differences in the figure. Remember that the figures must be self-explanatory so that the reader can identify the differences. Indicate them with symbols and present these symbols in the figure caption. Follow these same guidelines for the other figures presented in the text.

Discussion:

22- The discussion is the place to explain the results found in the study and not to repeat results. Please adjust.

23- Try to use manuscripts that did similar research to explain the findings of the present work. Follow a temporal linearity to explain the findings as presented in the methodology and results.

24- At the end of the discussion, present the limitations of the study.

Conclusions:

25- Focus on completing the study objectives without speculating information and without citing articles. Remember, the conclusions are from this article and not from others.

Author Response

(The authors gave the same response as above.)

Round 2

Reviewer 1 Report

Comments and Suggestions for Authors

Thank you for revision the paper. Please explain the abbreviation 

𝑅𝑀𝑆 

and include it in the table with all the abbreviations. 

Reviewer 2 Report

Comments and Suggestions for Authors

Dear authors,

Thank you for providing the revised version of the manuscript. It was verified that most of the comments were addressed and that the authors justified those that were not revised. Therefore, I believe that the manuscript can continue on the path to approval.